# Does Recurrence of Carpal Tunnel Syndrome (CTS) after Complete Division of the Transverse Ligament Really Exist?

**DOI:** 10.3390/jcm10184208

**Published:** 2021-09-17

**Authors:** Fatma Kilinc, Bedjan Behmanesh, Volker Seifert, Gerhard Marquardt

**Affiliations:** Department of Neurosurgery, Goethe-University, 60528 Frankfurt am Main, Germany; Bedjan.Behmanesh@kgu.de (B.B.); volker.seifert@kgu.de (V.S.); gerhard.marquardt@kgu.de (G.M.)

**Keywords:** carpal tunnel syndrome, revision surgery, peripheral nerve

## Abstract

The aim of this study was to evaluate whether recurrent carpal tunnel syndrome (CTS) after complete and sufficient division of the transverse ligament really exists. Another goal was to analyze the underlying reasons for recurrent CTS operated on in our department. Over an observation period of eleven years, 156 patients underwent surgical intervention due to CTS. The records of each patient were analyzed with respect to baseline data (age, gender, affected hand), as were clinical signs and symptoms pre- and postoperatively. To assess long-term results, standardized telephone interviews were performed using a structured questionnaire in which the patients were questioned about persisting symptoms, if any. Of the 156 patients, 128 underwent first surgical intervention due to CTS in our department. In long-term follow-up, two-thirds of these patients had no symptoms at all; one-third of the patients described mild persisting numbness. None of the patients experienced a recurrence of CTS. The 28 patients who received their first operation outside of our department were operated on for recurrent CTS. The cause of recurrence was incomplete division of the distal part of the transverse carpal ligament in all cases. The results suggest that recurrent CTS after complete and sufficient division of the transverse ligament is very unlikely.

## 1. Introduction

Carpal tunnel syndrome (CTS) is the most common peripheral nerve entrapment neuropathy [1]. The prevalence of CTS is approximately 5% in the general population [2,3]. It causes pain, numbness and/or tingling in the hand and arm and disturbed motor function of the hand [4,5]. In addition to sensory impairments (e.g., numbness and paresthesias), motor deficits occur in intrinsic median-innervated muscels such as the abductor pollicis brevis, the superficial belly of the flexor pollicis brevis and opponens pollicis. Compared to men, women are more likely to suffer from CTS [6]. Mild symptoms may be treated conservatively; however, if symptoms persist or increase, surgical treatment is required [5]. Nearly 20% of all patients with CTS are subjected to surgery [7]. Current operative measures include open, mini-open, and endoscopic decompression of the median nerve at the wrist [1,8,9].

The aim of all three operating techniques is to completely sever the transverse carpal ligament and, thus, to decompress the median nerve within the carpal tunnel [10,11,12].

In 3% to 25% of the patients treated surgically, however, persistence or recurrence of symptoms is encountered [10]. Assumed reasons for failure of surgical treatment with persisting or recurrent CTS are the incomplete transection of the transverse carpal ligament or the formation of scar tissue.

Carpal tunnel syndrome (CTS) may be a nonspecific manifestation of hereditary ATTR amyloidosis (ATTRm). Amyloid deposits in the transverse carpal ligament have also been observed in recurrent CTS [13,14,15]. Anatomic innovation variants such as the median-ulnar junction (MUC) branch, also known as the Martin-Gruber anastomosis, can lead to interpretation errors in nerve conduction studies of patients with carpal tunnel syndrome [16]. As early as 1998, Rempel et al. stated that case definitions that included electrodiagnostic examination results are of a higher specificity than case definitions that did not include electrodiagnostic examinations. Rempel et al. evaluated symptoms and an appropriate electrodiagnostic examination. If no electrodiagnostic examination was available, it was evaluated on the basis of the symptoms and physical examination findings. Based on the results, these were classified as “classic/probable”, “possible”, or “unlikely” [17].

The goals of this study were, therefore, to scrutinize whether recrudescence of CTS really exists in long-term follow-up after complete transection of the transverse ligament. Another goal was to analyze the underlying reasons for recurrent CTS operated on in our department.

## 2. Patients and Methods

The clinical data of 156 consecutive patients who underwent operations for CTS between January 2008 and February 2020 at our institution were retrospectively obtained by reviewing medical reports and outpatient charts. The records of each patient were analyzed with respect to baseline data (age, gender, affected hand) and clinical signs and symptoms. All included patients had a confirmatory electrophysiological examination before surgery. Comparing the baseline characteristics, in both groups, the majority of patients were female and the right hand was mostly affected. The recurrent carpal tunnel syndrome patients were included if they described persisting or recurring symptoms such as numbness, night pain, or weakness after previous open release with a confirming positive electromyography (EMG). However, data on the severity of EMG findings were not available because a large proportion of patients had them performed outside of our department and an accurate classification was lacking. The presence of amyloid deposition as a possible cause of recurrent CTS, as well as anatomic innovation variants such as the median-ulnar communicating branch, was not explicitly investigated in this study.

Recurrent CTS was defined as the recurrence of symptoms after a symptom-free interval of more than 3 months following surgery. When symptoms persisted directly after surgery or came back within 3 months after surgery, it was defined as persistent.

The primary operations were performed by, or under assistance of, the senior author. In the event of primary surgery (*n* = 128), all patients were operated on under regional anesthesia using the open mini-incision technique.

In doing so, care was taken that the transverse ligament was transected completely. Patients with persisting or recurrent CTS (*n* = 28) were operated on under general anesthesia. They were treated using the open technique of carpal tunnel decompression. For the surgical procedure, patients were positioned, as in the first surgery, on the operating table, and the arm was placed in 90° abduction on the surgical table. The existing scar was opened and the nerve was released from the superficial scar tissue. To protect the median nerve intraoperatively, a dissector was used. In all cases, the transverse ligament was fully visualized. The ligament was completely incised and a dissector was used to check whether the decompression was enough or not. If necessary, an operating microscope was used. A hypothenar fat pad flap, which is described in the literature as a reliable solution for recurrent CTS, was, at the request of the patients, not used [3,5,18,19].

All patients were discharged on the day of surgery and were suggested a follow-up visit three months postoperatively. To evaluate long-term results, standardised telephone interviews were conducted using a structured, one-time, detailed questionnaire in which patients were asked about any persisting symptoms. This questionnaire was designed for this study only. Patients answered the questions on symptoms with “yes” or “no”.

## 3. Results

### 3.1. First Operation for CTS

Of the 128 patients who had undergone first surgical intervention due to CTS in our department, follow-up data 3 months postoperation were available for only 98 patients because 30 patients skipped the follow-up evaluation. Preoperatively, all patients described one or more complaints, which included nocturnal pain, numbness, weakness, and/or pain increasing with activity over the median nerve distribution area. Three months postoperation, none of the patients had nocturnal pain or palsies, numbness had decreased significantly, and two-thirds of the patients were completely asymptomatic.

In order to conduct the patient survey for the assessment of long-term results as comprehensively and completely as possible, all patients were contacted by phone. Fourteen patients refused to participate in this study or could not be located; three patients had since deceased. Thus, a total of 111 complete telephone interviews, accounting for a respondent rate of 86.7%, were conducted. In doing so, 10 patients were contacted more than 10 years after surgery, 15 patients more than eight years postoperation, 21 patients more than five years after surgery, and 82 patients between one and four years postoperation (mean follow-up time—4.6 ± 3.3). Of these, 62 of the patients (55.9%) were female and 49 (44.1%) were male. Mean age was 57 years (ranging from 36 to 86). Twelve of these patients had both sides operated, so 123 decompression procedures of the median nerve were performed on 111 patients. Consequently, 67 (60.4%) operations were performed for the right hand and 56 (39.6%) for the left hand. Baseline data are shown in Table 1, and clinical data are depicted in Table 2. Lately, 76 of these 111 patients (68.5%) had no symptoms at all and 35 patients (31.5%) described mild symptoms, especially numbness. All these patients had described numbness preoperatively (Table 2), and it was still present directly after surgery, but much milder. Whether there was a correlation with the severity of CTS could not be determined as there was no data on severity from EMGs performed outside our department. None of the patients experienced a recurrence of CTS.

### 3.2. Operation for Persisting or Recurrent CTS

In all 28 cases, the first surgery was not performed in our hospital, and patients were assigned due to persistent or recurring symptoms. Sixteen of the patients (57.1%) were female and twelve (43.9%) were male. The mean age was 60 years (ranging from 36 to 86). Twenty-two (78.6%) operations were performed for the right hand and six (21.4%) for the left hand. Patients’ characteristics are shown in Table 3.

Preoperatively, all patients described one or more complaints, which included nocturnal pain, numbness, weakness and/or pain increasing with activity over the median nerve distribution area. The time interval between first and revision surgeries ranged from one to 312 months.

When performing revision surgery, an incomplete transection of the distal part of the transverse ligament was encountered in 19 patients; in nine cases, the transverse ligament was completely intact. A perineural fibrosis in any recurrence was not detected.

Three months postoperation, 17 patients described no symptoms after revision surgery; 10 patients described an improvement in numbness; one patient described no improvement after revision surgery. The clinical data are depicted in Table 4.

## 4. Discussion

The aim of surgical treatment of CTS is to completely sever the transverse carpal ligament in order to decompress the canal and the median nerve. For this purpose, several surgical techniques are at hand, including open and closed techniques [11,13,20,21]. In our study, all revision surgeries were performed with open techniques.

Recurrent or persisting CTS is thought to be due to an incomplete transection of the transverse ligament. Other authors suggested postoperative complications such as scarring, cystic formations, or postoperative adhesions to be the cause of recurrent CTS [12,13,22,23]. Stütz et al. described that, in 23% of the included patients, scar tissue was a reason for recurrent symptoms which caused a constriction of the nerve. In our 28 cases with recurrent CTS, however, we regularly found only superficial scar tissue, which was not the reason for recurrent symptoms [15]. The intraoperative appearance of the retinaculum was shown as a reason for recurrence or persistence of an insufficient decompression in the course of the first operation. In 19 patients, the transverse ligament was severed incompletely, and in nine patients, it was intact entirely so that the median nerve was still compressed. Some surgeons describe a reformation of the ligament after transection. In our study, 53% of the included recurrent patients had a symptom-free interval after primary surgery of less than one month, so a reformation cannot be assumed [15,22]. Nevertheless, it should be mentioned that in one case of reoperation, a symptom-free interval of 300 months was observed. After revision surgery with an intraoperative finding of an intact tunnel, a reformation of the ligament is most likely to be assumed here. In seven of the nine cases of an intraoperative finding of an intact tunnel during revision surgery, no symptoms were described for at least 3 months after follow-up. A reason for this could also be that the ligament was not manipulated during the first operation and thus there was no scarring with an additional pressure on the nerve. With a completely incised ligament, recovery of the median nerve was possible.

Carpal tunnel syndrome (CTS) may also be a nonspecific manifestation of hereditary ATTR amyloidosis (ATTRm). Recent studies described that amyloid is present in most biopsies from the transverse carpal ligament of such patients. Amyloid deposits in the transverse carpal ligament have also been observed in recurrent CTS. However, this study did not investigate whether recurrent CTS was caused by hereditary amyloidosis [14,24].

Sun et al. reported improvement of symptoms and function after revision surgery. They included 112 patients and both surgeries; primary as well as revision surgeries were performed by open technique. Patients with intraoperative completely intact ligaments were not described. An incomplete release of the transverse carpal ligament or incorrect diagnosis were described as the main causes of persistent symptoms. In 32% of cases, a complete relief was described [25]. Jones et al. described a complete relief of symptoms after open release in 57% of cases and 56% after endoscopic carpal tunnel release [22]. We reported that 61% of participants experienced no symptoms after revision surgery. A reason for this may be that the majority of symptom-free patients after revision surgery had an incomplete ligament intraoperatively, which meant that the median nerve was not compressed along the entire length of the ligament, but rather, as was shown during surgery, in the area of the distal end. It can be assumed that decompression of the distal ligament resulted in rapid improvement of symptoms.

Nevertheless, 10 of the 28 patients were free of symptoms for more than 48 months, and six of these even had a symptom-free interval of more than 120 months. A reason for this might be that proximal transection of the transverse ligament led to a temporary decompression of the median nerve and symptoms recurred as the distal part of the ligament thickened continuously in the natural course of the disease, thus resulting in a new compression of the nerve.

Stütz et al. described only two cases in a total of 2350 patients that needed revision after primary carpal tunnel decompression in their department [15]. Of the 128 patients who received first surgery at our institution, long-term follow-up data are available from 111 patients. Two-thirds of these patients had no symptoms at all years after surgery, while one-third of the patients described mild symptoms by means of persisting numbness. None of the patients experienced a recurrence of CTS.

## 5. Limitations

The present study is limited in various aspects. Acquisition of data was carried out retrospectively. The included patients with recurrent CTS were operated by just one surgeon, meaning there was a high risk for selection bias. We know that this study is limited due to the small sample group, but we are still following all our patients in order to obtain a large series. Nevertheless, the results of this study and further postoperative periods of time will need to be clarified in further studies.

## 6. Conclusions

In summary, to our knowledge, this is the very first long-term study to suggest that recurrent CTS is very unlikely after complete and sufficient transection of the transverse ligament during the first surgery. For persistent or recurrent CTS, the cause may be considered to be inadequate decompression at the time of initial surgery.

## Figures and Tables

**Table 1 jcm-10-04208-t001:** Patients’ characteristics for first surgery.

Patients Characteristics	
Parameter	Baseline characteristics (*n* = 111)
Age (yr)	67 (28–92)
Sex	
Male	49 (44.1)
Female	62 (55.9)
Operated hand	
Right	67 (60.4)
Left	56 (39.6)

**Table 2 jcm-10-04208-t002:** Symptoms and signs of patients preoperatively and at 3 months postoperatively (*n* = 111 patients/123 hands) after first surgery and revision surgery (*n* = 28 patients/28 hands).

	Pre- and Postoperation *n* (%) First Surgery	Pre- and Postoperation *n* (%) Revision Surgery
Night pain	38 (30.9%) 0	22 (78.6%) 0
Numbness	113 (91.9%) 35 (28.5%)	24 (85.7%) 11 (39.3%)
Weakness	85 (69.1%) 0	11 (39.2%) 1 (3.6%)
Tinel’s sign	52 (42.3%) 0	12 (42.9%) 0
No symptoms after surgery	83 (67.5%)	17 (60.7%)

**Table 3 jcm-10-04208-t003:** Patients’ characteristics for revision surgery.

Parameter	Baseline Characteristics (*n* = 28)
Age (yr)	60 (36–86)
Sex	
Male	12 (42.8)
Female	16 (57.1)
Operated hand	
Right	22 (78.6)
Left	6 (21.4)

**Table 4 jcm-10-04208-t004:** Postoperative outcomes in 28 revision surgery cases.

n	Age (Years)	Sex (F/M)	Operated Hand (R/L)	Symptoms before Recurrence Surgery	Symptom-Free Interval (Month)	Delay (Month)	Intraoperative Retinaculum Appearance	Follow-Up Month	Symptoms after Recurrence Surgery
1	52	M	R	H/P	0	24	incomplete	3	H
2	53	F	R	H/P/T/Pa	3	6	intact	3	H
3	53	F	R	H/A/Pa	9	14	incomplete	3	H
4	76	M	R	H/P/Pa	0	6	incomplete	3	H
5	58	F	L	H/P/Pa/T	0	9	intact	3	no symp
6	49	F	R	H/P/Pa/T	0	1	incomplete	No	no symp
7	51	F	R	H/Pa	0	14	incomplete	3	no symp
8	45	F	R	P	300	312	intact	No	no symp
9	57	M	R	H/Pa/A/T	0	8	incomplete	3/6	H/A
10	79	M	R	H/Pa/A/P	0	5	incomplete	No	no smyp.
11	67	M	R	H/Pa/T/P	0	14	incomplete	3	H
12	66	F	L	H/Pa	132	144	incomplete	3	no symp
13	45	F	R	H/Pa/T	0	25	intact	3	no symp
14	62	M	R	H	10	11	intact	No	no symp
15	62	F	R	H/Pa/T	0	10	intact	No	no symp
16	56	M	R	H/Pa/P	240	246	incomplete	4	no symp
17	60	F	R	Pa	120	204	incomplete	6	no symp
18	86	F	R	H	0	2	incomplete	3/9	no symp
19	36	F	R	H/P/A/T	1	6	incomplete	3	H
20	60	F	L	H/Pa/T	48	55	incomplete	No	H
21	79	M	L	H/Pa/T	0	5	intact	3	H
22	38	M	R	Pa/H/P	204	212	incomplete	3	no symp
23	59	F	R	Pa/H/P	92	98	incomplete	No	no symp
24	56	M	R	Pa/H/P	60	78	incomplete	No	no symp
25	83	F	R	H/Pa/P/T	0	5	incomplete	3	H
26	74	M	L	H/Pa	0	24	intact	3	no symp
27	50	F	R	H/P/T	1	5	incomplete	3	Same
28	61	F	L	H/Pa/T	17	20	intact	3	no symp

H = hypoesthesia, P = paresis, A = atrophy, Pa = pain, T = Tinnel sign, Delay = time elapsed between primary and revision surgery.

## Data Availability

The raw data supporting the conclusions of this article will be made available by the authors, without undue reservation.

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
