# Peer review of "Does Recurrence of Carpal Tunnel Syndrome (CTS) after Complete Division of the Transverse Ligament Really Exist?"

_jcm, 2021, doi:10.3390/jcm10184208_

Round 1
Reviewer 1 Report
The study explores assesses the recurrence rate of carpal tunnel syndrome (CTS) after complete division of the transverse ligament. None of the patients experienced a recurrence of CTS, suggesting that recurrent CTS after complete and sufficient division of the transverse ligament does not exist.
The study sample is adequate. The design and statistical analysis are good. However, I have some concerns about the neurophysiological evaluation performed.
-The reference list should be revised. I suggest to add relevant studies on CTS in the introduction, i.e. “Padua L, et al. Carpal tunnel syndrome: clinical features, diagnosis, and management. Lancet Neurol. 2016 Nov;15(12):1273-1284. doi: 10.1016/S1474-4422(16)30231-9. Epub 2016 Oct 11. PMID: 27751557.”.
I have some further suggestions in the next points on references.
-The authors should mention the technique performed and classification used for nerve conduction studies and CTS severity assessment (i.e. “Rempel D, et al. Consensus criteria for the classification of carpal tunnel syndrome in epidemiologic studies. Am. J. Public Health 1998, 88, 1447–1451.”).
-The authors stated: “All included patients had a confirmatory electrophysiological examination before surgery”. However, there is a lack of stratification of results depending on CTS severity assessed by means of nerve conduction studies. Also, there is no mention regarding median to ulnar anastomoses in the upper limb. In fact, Martin-Gruber anastomoses may lead to interpreting errors during routine nerve conduction studies in patients with CTS. In the presence of CTS, the CMAP at elbow could present an initial positive deflection thus leading to an apparently fast nerve conduction velocity of the median nerve in the forearm. These alterations are not always easy to recognize, despite marked discrepancy between clinical and electrodiagnostic findings. Hence, failure in diagnosis can result in an underrating of CTS severity. See also “Di Stefano V, et al. Median-to-Ulnar Nerve Communication in Carpal Tunnel Syndrome: An Electrophysiological Study. Neurol Int. 2021 Jul 12;13(3):304-314. doi: 10.3390/neurolint13030031. PMID: 34287351; PMCID: PMC8293426.”
-Did the author explored the possibility that recurrence of CTS might be caused by hereditary amyloidosis (ATTR)? In fact, CTS is a typical feature of ATTR, especially in males with recurrence of CTS. It was reported that CTS is the first sign of the disease in 60% of ATTR patients. See and discuss also: “Russo M, et al. Use of Drugs for ATTRv Amyloidosis in the Real World: How Therapy Is Changing Survival in a Non-Endemic Area. Brain Sci. 2021 Apr 27;11(5):545. doi: 10.3390/brainsci11050545. PMID: 33925301; PMCID: PMC8146901.”.
Author Response
Point 1: The reference list should be revised. I suggest to add relevant studies on CTS in the introduction, i.e. “Padua L, et al. Carpal tunnel syndrome: clinical features, diagnosis, and management. Lancet Neurol. 2016 Nov;15(12):1273-1284. doi: 10.1016/S1474-4422(16)30231-9. Epub 2016 Oct 11. PMID: 27751557.”.
Response 1: Carpal tunnel syndrome (CTS) is the most common peripheral nerve entrapment neuropathy.14
- Padua, Luca; Coraci, Daniele; Erra, Carmen; Pazzaglia, Costanza; Paolasso, Ilaria; Loreti, Claudia; Caliandro, Pietro; Hobson-Webb, Lisa D (2016). Carpal tunnel syndrome: clinical features, diagnosis, and management. The Lancet Neurology, 15(12), 1273–1284.
Point 2: The authors should mention the technique performed and classification used for nerve conduction studies and CTS severity assessment (i.e. “Rempel D, et al. Consensus criteria for the classification of carpal tunnel syndrome in epidemiologic studies. Am. J. Public Health 1998, 88, 1447–1451.”).
Response: Already in 1998 Rempel et al. described that case definitions that included electrodiagnostic examination results a higher specificity than case definitions that did not include electrodiagnostic examinations. Rempel et al. evaluated symptoms and an appropriate electrodiagnostic examination. If no electrodiagnostic examination was available, it was evaluated on the basis of the symptoms and physical examination findings. Based on the results, these were classified as "classic/probable," "possible," or "unlikely." 15
- Rempel D, et al. Consensus criteria for the classification of carpal tunnel syndrome in epidemiologic studies. Am. J. Public Health 1998, 88, 1447–1451.”).
Point 3: -The authors stated: “All included patients had a confirmatory electrophysiological examination before surgery”. However, there is a lack of stratification of results depending on CTS severity assessed by means of nerve conduction studies. Also, there is no mention regarding median to ulnar anastomoses in the upper limb. In fact, Martin-Gruber anastomoses may lead to interpreting errors during routine nerve conduction studies in patients with CTS. In the presence of CTS, the CMAP at elbow could present an initial positive deflection thus leading to an apparently fast nerve conduction velocity of the median nerve in the forearm. These alterations are not always easy to recognize, despite marked discrepancy between clinical and electrodiagnostic findings. Hence, failure in diagnosis can result in an underrating of CTS severity. See also “Di Stefano V, et al. Median-to-Ulnar Nerve Communication in Carpal Tunnel Syndrome: An Electrophysiological Study. Neurol Int. 2021 Jul 12;13(3):304-314. doi: 10.3390/neurolint13030031. PMID: 34287351; PMCID: PMC8293426.”
Response 3:
(Introduction) Anatomic innovation variants such as the median-ulnar junction (MUC) branch, also known as the Martin-Gruber anastomosis, can lead to interpretation errors in nerve conduction studies in patients with carpal tunnel syndrome. 4
- Di Stefano V, et al. Median-to-Ulnar Nerve Communication in Carpal Tunnel Syndrome: An Electrophysiological Study. Neurol Int. 2021 Jul 12;13(3):304-314.
(Methods) However, data on the severity of EMG findings were not available because a large proportion of patients had them performed outside our department and an accurate classification was lacking.
The presence of amyloid deposition as a possible cause of recurrent CTS as well as anatomic innovation variants like the median-ulnar communicating branch was not explicitly investigated in this study.
Point 4: -Did the author explored the possibility that recurrence of CTS might be caused by hereditary amyloidosis (ATTR)? In fact, CTS is a typical feature of ATTR, especially in males with recurrence of CTS. It was reported that CTS is the first sign of the disease in 60% of ATTR patients. See and discuss also: “Russo M, et al. Use of Drugs for ATTRv Amyloidosis in the Real World: How Therapy Is Changing Survival in a Non-Endemic Area. Brain Sci. 2021 Apr 27;11(5):545. doi: 10.3390/brainsci11050545. PMID: 33925301; PMCID: PMC8146901.”.
Response 4:
(Introduction) Carpal tunnel syndrome (CTS) may also be a nonspecific manifestation of hereditary ATTR amyloidosis (ATTRm). Amyloid deposits in the transverse carpal ligament have also been observed in recurrent CTS.11,16,20
(Methods) The presence of amyloid deposition as a possible cause of recurrent CTS as well as anatomic innovation variants like the median-ulnar communicating branch was not explicitly investigated in this study.
Reviewer 2 Report
Congratulations for the manuscript.
CTS is always an interesting item to study because of its high prevalence.
I consider your text need a revision, I could give you some advices:
Introduction:
- Specify that the motor alterations due to the carpal tunnel syndrome are related to the intrinsic musculature of the thumb.
- no reference is made to other causes of CTS recurrence
Methods:
- inclusion bias: define recurrence as new cases + persistent symptoms; differentiate both of them
- In none of the revision surgeries required a technique to protect the median nerve, Was there really no perineural fibrosis in any recurrence?
Results:
- how do you calculate the "significant decrease" in numbness measured in the postoperative period?
- 31.5% of patients who describe mild symptoms after surgery, do these symptoms start in the immediate postoperative period or are currently new? Does it relate to the severity of the CTS measured by electromiography test?
- how would you explain the case of reoperation after 312 months, with an intraoperative finding of an "intact" tunnel, who recovers 100% in the current postoperative period?
Discussion:
- Many causes of CTS recurrence are described in the literature, not only incomplete release, and the appearance of fibrosis as an intraoperative finding in revision surgery is very significant, a finding not found in this series. How do the authors explain the symptom-free interval greater than one month and less than 2 years?
- The authors justify the reason for the improvement in the postoperative period of their review cases, as they are patients with incomplete release, however 7 of the 9 with "intact" tunnel also have no symptoms. How do they explain it?
Author Response
Point 1: Specify that the motor alterations due to the carpal tunnel syndrome are related to the intrinsic musculature of the thumb.
Response 1: In addition to sensory impairments (e.g., numbness and paresthesias), motor deficits in the abductor pollicis brevis, the superficial belly of the flexor pollicis brevis and opponens pollicis, which are the intrinsic median-innervated muscles also occure.
Point 2: no reference is made to other causes of CTS recurrence
Response 2: Carpal tunnel syndrome (CTS) may be a nonspecific manifestation of hereditary ATTR amyloidosis (ATTRm). Amyloid deposits in the transverse carpal ligament have also been observed in recurrent CTS.11,16,20
Anatomic innovation variants such as the median-ulnar junction (MUC) branch, also known as the Martin-Gruber anastomosis, can lead to interpretation errors in nerve conduction studies in patients with carpal tunnel syndrome. 4
- Di Stefano V, et al. Median-to-Ulnar Nerve Communication in Carpal Tunnel Syndrome: An Electrophysiological Study. Neurol Int. 2021 Jul 12;13(3):304-314.
- Milandri, A.; Farioli, A.; Gagliardi, C.; Longhi, S.; Salvi, F.; Curti, S.; Foffi, S.; Caponetti, A.G.; Lorenzini, M.; Ferlini, A.; et al. Carpal Tunnel Syndrome in Cardiac Amyloidosis: Implications for Early Diagnosis and Prognostic Role across the Spectrum of Aetiologies. Eur. J. Heart Fail. 2020, 22, 507–515.
- Russo M, et al. Use of Drugs for ATTRv Amyloidosis in the Real World: How Therapy Is Changing Survival in a Non-Endemic Area. Brain Sci. 2021 Apr 27;11(5):545.
- Stütz N, Gohritz A, van Schoonhoven J, Lanz U. Revision surgery after carpal tunnel release--analysis of the pathology in 200 cases during a 2 year period. J Hand Surg Br. 2006 Feb;31(1):68-71.
Point 3: - inclusion bias: define recurrence as new cases + persistent symptoms; differentiate both of them
Response 3:
Recurrent CTS was defined as the recurrence of symptoms after a symptom-free interval more than 3 month following surgery. When symptoms persisted directly after surgery or came back within 3 months after surgery, it was defined as persistent.
Point4: In none of the revision surgeries required a technique to protect the median nerve, Was there really no perineural fibrosis in any recurrence?
Response 4: . For protecting the median nerve intraoperatively a dissector was used. . In all cases the transverse ligament was fully visualized. The ligament was completely incised and a dissector was used to check whether the decompression was enough or not. If necessary, an operating microscope was used.
A perineural fibrosis in any recurrence was not detected.
Point 5: how do you calculate the "significant decrease" in numbness measured in the postoperative period?
Response 5: Lately, 76 of these 111 patients (68.5%) had no symptoms at all, 35 patients (31.5%) described mild symptoms, especially numbness.
Point6: 31.5% of patients who describe mild symptoms after surgery, do these symptoms start in the immediate postoperative period or are currently new? Does it relate to the severity of the CTS measured by electromiography test?
Response 6: Lately, 76 of these 111 patients (68.5%) had no symptoms at all, 35 patients (31.5%) described mild symptoms, especially numbness. All these patients described numbness already preoperatively (Table 2), which was still present directly after surgery, but much milder. Whether there was an association with the severity of CTS measured by the electromiography test could not be determined because of a lack of data on the severity of EMG findings which was performed outside our department.
Point 7: how would you explain the case of reoperation after 312 months, with an intraoperative finding of an "intact" tunnel, who recovers 100% in the current postoperative period?
Response 7:
Nevertheless, it should be mentioned that in one case of reoperation a symptom-free interval of 300 month was observed. After revision surgery with an intraoperative finding of an intact tunnel a reformation of the ligament is most likely to be assumed here.
In 7 of 9 cases of intraoperative finding of an intact tunnel during revision surgery no symptoms after at least 3 months follow-up was described. A reason for this can also be, that the ligament was not manipulated during the first operation and thus there was no scarring with additional pressure effect on the nerve. After completely incised ligament a recovery of median nerve was possible.
Point 8: - Many causes of CTS recurrence are described in the literature, not only incomplete release, and the appearance of fibrosis as an intraoperative finding in revision surgery is very significant, a finding not found in this series. How do the authors explain the symptom-free interval greater than one month and less than 2 years?J Neurosurg. 2019 Feb 15;132(3):847-855.
Response 8:
A reason for this may be that the majority of symptom-free patients after revision surgery, had intraoperatively an incomplete ligament, which meant that the median nerve was not compressed along the entire length of the ligament, but rather, as was shown during surgery, in the area of the distal end. It can be assumed that decompression of the distal ligament resulted in rapid improvement of symptoms.
Nevertheless 10 of the 28 patients were free of symptoms for more than 48 months and six of these even had a symptom free interval of more than 120 months. A reason for this might be that proximal transection of the transverse ligament led to a temporary decompression of the median nerve and symptoms recurred as the distal part of the ligament thickened continuously in the natural course of the disease, thus resulting in a new compression of the nerve.
Point 9: The authors justify the reason for the improvement in the postoperative period of their review cases, as they are patients with incomplete release, however 7 of the 9 with "intact" tunnel also have no symptoms. How do they explain it?
Response 9:
In 7 of 9 cases of intraoperative finding of an intact tunnel during revision surgery no symptoms after at least 3 months follow-up was described. A reason for this can also be, that the ligament was not manipulated during the first operation and thus there was no scarring with additional pressure effect on the nerve. After completely incised ligament a recovery of median nerve was possible.
Reviewer 3 Report
Reviewer comments to the authors on manuscript “Does recurrence of carpal tunnel syndrome (CTS) after complete division of the transverse ligament really exist?”
General comments:
The study question is very important and this area lacks knowledge. The study design is acceptable. Mainly, the manuscript is logical and easy to read, but the results section and tables need to be revised for clarity, and discussion needs a little more explanation/narration.
If the authors plan on continuing research on this area, I think a prospective study design with several follow-up appointments/interviews with validated patient reported outcome measures and maybe ENMG for reliable comparison of outcomes would be even better. With this comment I’d like to encourage the authors to continue with future studies.
Introduction:
- row 26: “The prevalence of CTS is approximately 5%.” This sentence needs a reference and precision of the population it refers to (I assume that the authors mean general population).
Patients and methods:
- rows 46-47: “All included patients had a confirmatory electrophysiological examination before surgery.” This is mandatory in my opinion. Do the authors have data on the severity of ENMG findings? Usually, with patients with more severe ENMG findings some symptoms might remain for months, and with most severe cases, even permanently.
- rows 66-68: “To assess long-term results standardized telephone interviews were performed lately using a structured, one-time detailed questionnaire in which the patients were questioned about persisting symptoms, if any. “ Was this questionnaire designed for this study only? Did the patients answer the questions on symptoms yes/no or on a Likert scale?
Results:
- A comparison of outcomes of primary and revision surgery would improve the results section and make it more friendly for the reader. For example, Table 2 could be modified to include columns for primary and revision surgery results side to side.
- rows 82-83: “In doing so, 10 patients were contacted more than 10 years after surgery…” Mean follow-up time (sd) is needed here.
Discussion:
- rows 143 & 145: typing error, I think you mean scar tissue (not scare tissue).
- rows 153-154: “Sun and Soldani…” Could you describe these studies little more? How many patients did they include, was the primary/revision surgery performed by open technique? Did they have only patients with completely intact ligament in these studies (as you discuss about the incomplete division of the ligament further in the same paragraph)?
- rows 166-176: “Stütz et al. described only in two cases a needed revision after primary carpal tunnel decompression in their department.” Two patients out of how many?
Author Response
Point 1: row 26: “The prevalence of CTS is approximately 5%.” This sentence needs a reference and precision of the population it refers to (I assume that the authors mean general population).
Response 1: The prevalence of CTS is approximately 5% in the general population.7,12
- Hulkkonen S, Shiri R, Auvinen J, Miettunen J, Karppinen J, Ryhänen J. Risk factors of hospitalization for carpal tunnel syndrome among the general working population. Scand J Work Environ Health. 2020;46(1):43–9.
- Newington L, Harris EC, Walker-Bone K. Carpal tunnel syndrome and work. Best Pract Res Clin Rheumatol. 2015;29(3):440–53.
Point 2: rows 46-47: “All included patients had a confirmatory electrophysiological examination before surgery.” This is mandatory in my opinion. Do the authors have data on the severity of ENMG findings? Usually, with patients with more severe ENMG findings some symptoms might remain for months, and with most severe cases, even permanently.
Response 2: The recurrent carpal tunnel syndrome patients were included, if they described still persisting or recurred symptoms like numbness, night pain or weakness after previous open release with a confirming positive electromyography (EMG). However, data on the severity of EMG findings were not available because a large proportion of patients had them performed outside our department and an accurate classification was lacking.
Point 3: rows 66-68: “To assess long-term results standardized telephone interviews were performed lately using a structured, one-time detailed questionnaire in which the patients were questioned about persisting symptoms, if any. “ Was this questionnaire designed for this study only? Did the patients answer the questions on symptoms yes/no or on a Likert scale?
Response 3:
This questionnaire was designed for this study only. Patients answered the questions on symptoms with “yes” or “no”
Point 4: A comparison of outcomes of primary and revision surgery would improve the results section and make it more friendly for the reader. For example, Table 2 could be modified to include columns for primary and revision surgery results side to side.
Response 4:
See Table 2. Manuscript
Point 5: rows 82-83: “In doing so, 10 patients were contacted more than 10 years after surgery…” Mean follow-up time (sd) is needed here.
Response 5: In doing so, 10 patients were contacted more than 10 years after surgery, 15 patients more than eight years postoperatively, 21 patients more than five years after surgery and 82 patients between one and four years postoperatively (mean follow-up time 4,6±3,3).
Point 6: rows 143 & 145: typing error, I think you mean scar tissue (not scare tissue).
Response 6: . Stütz et al. described in 23% of the included patients as reason for recurrent symptoms scar tissue which causes a constriction of the nerve. In our 28 cases with recurrent CTS, however, we regularly found only superficial scar tissue which was not the reason for recurrent Symptoms.
Point 7: rows 153-154: “Sun and Soldani…” Could you describe these studies little more? How many patients did they include, was the primary/revision surgery performed by open technique? Did they have only patients with completely intact ligament in these studies (as you discuss about the incomplete division of the ligament further in the same paragraph)?
Response 7: "Soldani et al." was removed, ´"Jones et al." was introduced
Sun et al. reported improvement of symptoms and function after revision surgery. They included 112 patients and both surgeries, primary as well as revision surgery was performed by open technique. They don’t described patients with intraoperative finding with complete intact ligament. An incomplete release of the transverse carpal ligament or incorrect diagnosis were described as main causes of persistent symptoms. In 32% a complete relief was described. 21 Jones et al. described a complete relief of symptoms after open release in 57% and after endoscopic carpal tunnel release in 56%.9
- Jones NF, Ahn HC, Eo S. Revision surgery for persistent and recurrent carpal tunnel syndrome and for failed carpal tunnel release. Plast Reconstr Surg. 2012 Mar;129(3):683-692
- Sun PO, Selles RW, Jansen MC, Slijper HP, Ulrich DJO, Walbeehm ET. Recurrent and persistent carpal tunnel syndrome: predicting clinical outcome of revision surgery. J Neurosurg. 2019 Feb 15;132(3):847-855.
Point 8: rows 166-176: “Stütz et al. described only in two cases a needed revision after primary carpal tunnel decompression in their department.” Two patients out of how many?
Response 8: Stütz et al. described only in two cases in total of 2350 patients a needed revision after primary carpal tunnel decompression in their department.20
- Stütz N, Gohritz A, van Schoonhoven J, Lanz U. Revision surgery after carpal tunnel release--analysis of the pathology in 200 cases during a 2 year period. J Hand Surg Br. 2006 Feb;31(1):68-71.
Round 2
Reviewer 1 Report
I have no further suggestions.
Author Response
Thank you very much.
Reviewer 2 Report
Thank you for adapting the text to our corrections.
Even so, I consider that the first thing to be eliminated from the abstract is the last sentence in which you conclude that the recurrence of the CTS does not exist, because as the authors end up, showing in the corrections 3 and 7, there is a "reformation of the tunnel" in those patients who are free of symptoms 312, more than 200, or even 144 days and they have been reoperated with a finding of a physically closed tunnel. Are their tunnels incompletely freed or incompletely refurbished? In the end, the answers to some of my doubts about the absence of recurrences are answered with non-demonstrable assumptions, as in point 9.
Congratulations on your work.
Author Response
The results suggest that recurrent CTS after complete and sufficient division of the transverse ligament is very unlikely.
Thank you very much.